# Computationally Efficient Laplace Approximations for Neural Networks

**Swarnali Raha**
Department of Statistics
University of Florida
Gainsville, FL
swarnali.raha@ufl.edu

**Kshitij Khare**
Department of Statistics
University of Florida
Gainsville, FL
kdkhare@ufl.edu

**Rohit K Patra**
Linkedin Inc
New York, NY
ropatra@linkedin.com

## Abstract

Laplace approximation is arguably the simplest approach for uncertainty quantification using intractable posteriors associated with deep neural networks. While Laplace approximation based methods are widely studied, they are not computationally feasible due to the involved cost of inverting a (large) Hessian matrix. This has led to an emerging line of work which develops lower dimensional or sparse approximations for the Hessian. In this paper, we build upon this work by proposing two novel sparse approximations of the Hessian: (1) greedy subset selection, and (2) gradient based thresholding. We show via simulations that these methods perform well when compared to current benchmarks over a broad range of experimental settings.

## 1 Introduction

Uncertainty quantification for neural networks is critical in understanding the quality of output/estimates provided by the network, and is a crucial component for many standard frameworks (such as contextual bandits) to study exploration-exploitation trade-offs. Such frameworks are useful in many domains, such as recommender systems [Su et al., 2024, Joachims et al., 2018, Li et al., 2010], content moderation [Avadhanula et al., 2022], healthcare [Durand et al., 2018, Mintz et al., 2020, Esteva et al., 2017], dynamic pricing [Misra et al., 2019], dialogue systems [Liu et al., 2018], self-driving cars [Bojarski et al., 2016], detecting hallucinations in LLMs [Chen and Mueller, 2023, Felicioni et al., 2024, Dwaracherla et al., 2024]. It is well known that accurate uncertainty quantification can significantly improve the performance of downstream tasks [Ovadia et al., 2019, Riquelme et al., 2018].

Despite these benefits, uncertainty quantification is not yet widely used in deep learning due to excessively high computational cost in cutting edge applications. This is in particular true for the Laplace approximation based approach, which approximates the posterior distribution of the parameters in a neural network by a multivariate Gaussian centered at the posterior mode $\hat{\boldsymbol{\theta}}_{MAP}$ (with $\boldsymbol{\theta}$ denoting the parameters in the network), and with the (negative) observed Fisher information matrix as its precision (inverse-covariance) matrix, see [MacKay, 1992, Foong et al., 2019]. This approach is technically well-grounded and is also attractive due to its conceptual simplicity. However, any application of this approach for uncertainty quantification requires inversion of the observed Fisher information matrix whose computational complexity $O(p^3)$ can be prohibitive in many settings (here $p$ is the number of parameters in the neural network).

To overcome this computational challenge, a nascent strand of the literature has focused on replacing the Laplace precision matrix $\Omega$ by a lower dimensional approximation which retains the most relevant components or entries. In [Nilsen et al., 2022], the authors propose replacing the Laplace precision matrix by an approximation which uses only the top $k$ eigenpairs of $\Omega$ (identified using the Lanczos algorithm [Cullum and Willoughby, 2002]). However, the iterative Lanczos algorithm can in general suffer from numerical instability issues when using finite precision arithmetic, especially for large matrices [Chen and Trogdon, 2024]. The *PrecisionDiag* approach in [Riquelme

et al., 2018, Zhang et al., 2020] approximates $\Omega$ by setting all off-diagonal entries to zero. This leads to extremely efficient computation, but can suffer from poor statistical performance given that it ignores *all* cross-parameter correlations.

Our particular focus in this paper is a class of methods, termed *sub-network LA* in [Daxberger et al., 2021a], that focuses on identifying a small subset $S$ of parameters capturing the most variability. The matrix $[\Omega_{S,S}]^0$, constructed by setting all entries of $\Omega$ outside of the principal submatrix $\Omega_{S,S} := (\Omega_{rs})_{r \in S, s \in S}$ to be zero, is then used as an approximation for $\Omega$. Since the approximation is singular, its Moore-Penrose inverse is utilized for subsequent variance computations. This amounts to ignoring the variability in all other parameters except those in $S$. To the best of our knowledge, two methods for identifying $S$ have been proposed in the current literature. The *Neural-Linear* [Snoek et al., 2015, Riquelme et al., 2018] method chooses $S$ as the subset of parameters in the last layer of the neural network. However, the last layer of the network may not always contain the most relevant parameters in terms of capturing variability (see Section 4). The marginal variance based approach in [Daxberger et al., 2021a, Section 5] relies on an independence assumption (ignoring cross-correlations) for the posterior distribution of the parameters. For a given subset size $k$, this approach reduces to choosing the parameters corresponding to the $k$ smallest diagonal entries of $\Omega$.

In this paper, we propose two methods for selecting the parameter subset $S$ for subnetwork LA which build on these promising initial efforts while aiming to address their deficiencies through a more refined treatment. The first method, *Greedy-Laplace* pursues a greedy step-wise approach for selecting $S$. The method starts with the Laplace precision matrix $\Omega$, and at each step the parameter corresponding to the smallest diagonal entry in the current precision matrix is chosen and added to $S$, and the precision matrix is shrunk and updated by 'removing the effect' of this parameter. The second method, *Gradient-Laplace*, recognizes that the posterior predictive variance for a future observation also depends on the corresponding gradient vector (see (1)), and uses parameters with the highest average absolute gradient evaluated over a reference dataset to construct $S$. Detailed empirical evaluation based on accuracy of subnetwork choice (Section 4) and coverage of predictive credible intervals (Appendix D) is provided.

## 2 Setup

In this article, our central focus will be on the **linearized Laplace Approximation** (LLA) [Foong et al., 2019]. We consider a neural network with output $f_{\boldsymbol{\theta}}(\mathbf{x})$ for given input $\mathbf{x}$ and parameter vector $\boldsymbol{\theta} \in \mathbb{R}^p$. The network is trained using a given dataset $\mathcal{D}_{train} = \{(\mathbf{x}_n, y_n) : 1 \leq n \leq N_{train}\}$, where we assume, $y_n \sim N(f_{\boldsymbol{\theta}}(\mathbf{x}_n), \sigma_0^2)$. We place independent Gaussian priors on the weights and biases of the network. As in [Foong et al., 2019], under this setup, the LLA provides the following approximation of the posterior predictive distribution for a new observation $(\mathbf{x}^*, y^*)$,

$$p(y^*|\mathbf{x}^*, \mathcal{D}_{train}) \approx N(f_{\hat{\boldsymbol{\theta}}_{MAP}}(\mathbf{x}^*), \sigma_0^2 + g(\mathbf{x}^*)^T \Omega^{-1} g(\mathbf{x}^*)) \tag{1}$$

where, $g(\mathbf{x}^*) = \nabla_{\boldsymbol{\theta}} f_{\boldsymbol{\theta}}(\mathbf{x}^*)|_{\boldsymbol{\theta} = \hat{\boldsymbol{\theta}}_{MAP}}$, and $\Omega$ is the negative Hessian matrix, which in practice, is replaced by the Gauss-Newton matrix, as the Gauss-Newton matrix is guaranteed to be positive semi-definite. In particular, letting $\mathbf{v}$ denote the vector containing all prior variances, we set

$$\Omega = \frac{1}{\sigma_0^2} \sum_{n=1}^{N} g(\mathbf{x}_n) g(\mathbf{x}_n)^T + (diag(\mathbf{v}))^{-1}. \tag{2}$$

The posterior approximation for a classification problem will be similar but with the Gauss-Newton matrix replaced by the Generalized Gauss-Newton matrix, see Appendix C for more details.

## 3 Proposed Algorithms

Our goal is to develop novel and principled methods for choosing the subset $S$ for subnetwork LA, so that the corresponding approximation $[\Omega_{S,S}]^0$ serves as an effective surrogate for $\Omega$ for the variance computation in (1). We now describe two strategies for this purpose.

**The Greedy-Laplace Approach**. Note that the inverse of the $r^{th}$ diagonal entry of a precision matrix is equal to the conditional variance of the $r^{th}$ variable given all the other variables. Given

a user-specified subset-size $k$, we leverage this fact to sequentially identify the 'best' subset of $k$ parameters (i.e., the subset capturing the most variability) in a greedy fashion. We start with $\Omega^{(1)} = \Omega$ (as in (2) and (C.1)). In the $i^{th}$ step, the parameter with the minimum diagonal entry in $\Omega^{(i)}$ is chosen, and the marginal precision matrix of the remaining variables is calculated and set as $\Omega^{(i+1)}$ (needs $O((p-i)^2)$ computations, see Step 5 of Algorithm 1). In particular, $\Omega^{(i+1)}$ is a $(p-i+1) \times (p-i+1)$ matrix. We stop after the $k^{th}$ step. We refer to this procedure as the *Greedy-Laplace algorithm* (Algorithm 1), which has an overall computational complexity of $O(kp^2)$. The output of Algorithm 1 is an index set $S$ (of size $k$), and the surrogate $[\Omega_{S,S}]^0$ is obtained from $\Omega$ by setting all elements outside the principal sub-matrix corresponding to $S$ to zero. Recall that since $[\Omega_{S,S}]^0$ is singular, its Moore-Penrose inverse is used in (1) instead of $\Omega^{-1}$.

---

**Algorithm 1** Greedy-Laplace Algorithm

---

**Require:** Laplace Precision matrix $\Omega$ as in (2); user specified dimension $k$
1: Initialize $\Omega^{(1)} = \Omega$, collection of indices of selected parameters $S = \phi$
2: **for** $t = 1, 2, \ldots, k$ **do**
3:      Compute $j = \underset{i \in \{1,2,\cdots,p\} \setminus S}{argmin} (\Omega^{(t)})_{i,i}$
4:      Update $S = S \cup \{j\}$
5:      Construct $\Omega^{(t+1)}$ by removing row & column for $j^{th}$ variable in $\Omega^{(t)}$, and updating remaining entries: $\Omega^{(t+1)} = (\Omega^{(t)})_{-j,-j} - \frac{(\Omega^{(t)})_{-j,j}(\Omega^{(t)})_{j,-j}}{(\Omega^{(t)})_{j,j}}$
6: **end for**
7: Return S

---

**The Gradient-Laplace Approach**. Note that the variance term in (1) depends on both $\Omega$ and $g(\mathbf{x}^*)$. While Algorithm 1 exclusively uses the matrix $\Omega$ to identify the subset $S$ of the most relevant parameters, an alternative is to focus instead on the gradient $g$. In particular, this gradient-based approach, considers a reference data set $\mathcal{D}$ (either the training data $\mathcal{D}_{train}$ or a separate test data set $\mathcal{D}_{test}$), and computes the average absolute gradient $\bar{g} = \frac{1}{|\mathcal{D}|} \sum_{\mathbf{x} \in \mathcal{D}} |g(\mathbf{x})|$. Here, for any vector $\mathbf{u} \in \mathbb{R}^p$, $|\mathbf{u}| := (|u_j|)_{j=1}^p$. We now choose $S$ to be the set of indices corresponding to the largest $k$ entries of $\bar{g}$. We refer to this procedure as the Gradient-Laplace algorithm (Algorithm 2). Again, the Moore-Penrose inverse of $[\Omega_{S,S}]^0$ is used as a surrogate for $\Omega^{-1}$ in (1). Note that this methods requires an additional summary statistic of the gradients, which in the case of $\mathcal{D}_{train}$ has negligible overhead but in the case of $\mathcal{D}_{test}$ requires a backprop over a large enough sub-sample.

---

**Algorithm 2** Gradient-Laplace Algorithm

---

**Require:** $\hat{\boldsymbol{\theta}}_{MAP}$ from fitted model using $\mathcal{D}_{train}$; Reference data $\mathcal{D}$; user-specified dimension $k$
1: Compute the gradient $g(\mathbf{x})$ for all $\mathbf{x} \in \mathcal{D}$
2: Compute the average of the absolute gradients over all data points in $\mathcal{D}$, denote by $\bar{g}$
3: Select the $k$ parameters corresponding to the $k$ highest values in $\bar{g}$
4: Collect the respective indices in $S$. Return $S$.

---

Note here that, computing $\hat{\boldsymbol{\theta}}_{MAP}$ and the gradients $g(\mathbf{x})$ above is required irrespective of the approximation. Now, for a reference dataset $\mathcal{D}$ of size $N$, computing $\bar{g}$ and selecting $k$ indices corresponding to the $k$ highest values in $\bar{g}$ using the sorting approach requires $O(max\{Np, p\,logp\})$ computations. Selecting the $k \times k$ surrogate submatrix from $\Omega$ and obtaining the Moore-Penrose inverse requires another $O(k^3)$ computations, making the overall complexity $O(max\{Np, p\,logp, k^3\})$.

## 4 Empirical Evaluation

### 4.1 Experiment 1: Accuracy of Subnetwork Choice

**Experimental Setup**. [Daxberger et al., 2021b] propose using the Wasserstein distance between the linearized Laplace posterior distribution of the parameters $\boldsymbol{\theta}$ (multivariate normal with covariance matrix $\Omega^{-1}$) and its subnetwork LA based surrogate (singular multivariate normal with covariance

matrix given by the Moore-Penrose inverse of $[\Omega_{S,S}]^0$) as a measure of accuracy/quality of the sub-network based approximation using the subset $S$. We use a derived Wasserstein metric to compare the performance of the proposed methods (*Greedy-Laplace* and *Gradient-Laplace*) for choosing $S$ to existing subnetwork LA methods - the *Subnet diagonal* approach [Daxberger et al., 2021b, Section 5]), the *NeuralLinear* approach [Snoek et al., 2015, Riquelme et al., 2018] and the *Last k* approach (a natural extension of *NeuralLinear; see Appendix B*). For thoroughness, we also include the (non subnetwork LA based) *PresicionDiag* approach [Riquelme et al., 2018, Zhang et al., 2020] for obtaining a surrogate of $\Omega$. For further details on these benchmark methods see Appendix B.

For this experiment. two working model settings: mis-specified and well-specified, and three kinds of test data: in-distribution, out-of-distribution (OOD) and mixed domain are considered. For further details about the experimental settings and hyper-parameter choices see Appendix A. For each method, working model and test data combination, the following procedure is used to compute a Wasserstein distance based metric. For each test data observation, say $(\mathbf{x}^*, y^*)$, we compute the Wasserstein distance between the normal distribution in (1) and the normal distribution when $\Omega$ is replaced by its relevant surrogate. The average of these Wasserstein distances over the test dataset is then used as a measure of accuracy for the method in the given setting.

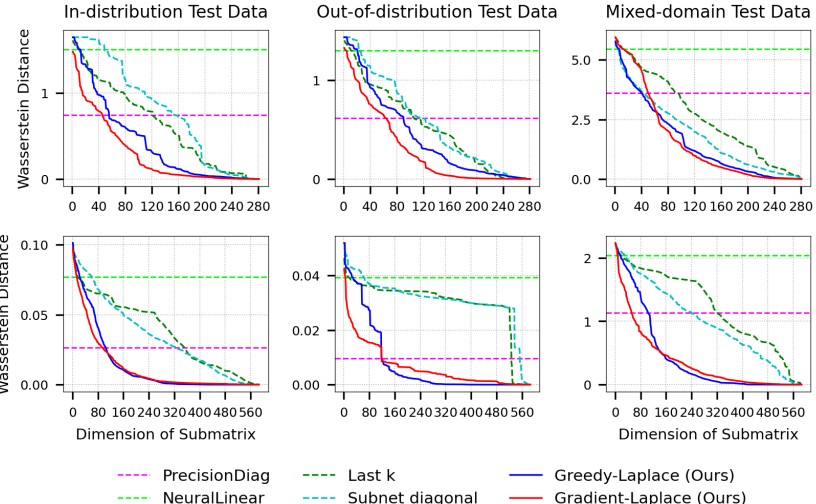

Figure 1: Wasserstein distances between the posterior predictive distribution using the full Laplace posterior and the relevant surrogate Laplace posterior, averaged over the entire test data set. The x-axis corresponds to the subset size $k$. The top row corresponds to the mis-specified working model setting, and the bottom row corresponds to the well-specified working model setting.

**Results and plots**. The plots in Figure 1 depict the (Wasserstein-based) accuracy measures for various methods under the six experimental settings discussed above. The plots show that both the proposed methods significantly and consistently provide superior approximations to the Laplace posterior compared to other subnetwork LA methods - the *Subnet diagonal* approach and the *Last k* approach - for each subset size $k$ and across all settings. The improvement in accuracy is remarkably more pronounced in the mis-specified working model setting compared to the well-specified working model setting. The accuracy of the *NeuralLinear* and *PrecisionDiag* approaches does not change with $k$, but we see that even for small or moderate values of $k$, the proposed methods start providing comparatively better performance. Finally, the performance of the two proposed methods is similar, with Gradient-Laplace providing a marginally better approximation overall. To conclude, these experiments strongly demonstrate that the proposed methods provide more effective and refined choices for computationally efficient approximations of the Laplace posterior.

## 4.2 Experiment 2: Coverage of Confidence Intervals

In this section, we compare the performance of our proposed methods with several benchmark methods using the coverage of posterior predictive credible intervals as a measure of performance. In particular, we consider two different settings for this experiment: *Regression* and *Classification*.

Two working models - a mis-specified working model and a well-specified working model are considered. For further details about the experimental settings see Appendix D. For each method (with each setting and working model), we compute the empirical coverage of 95% credible intervals constructed from the posterior predictive distribution for new observations over a test data set. In particular, for each future (test data) observation with predictor vector $\mathbf{x}^*$, we check if the corresponding mean response value $f_{\boldsymbol{\theta}_0}(\mathbf{x}^*)$ lies in the 95% posterior predictive credible interval. The empirical coverage is the proportion of such inclusions in the test dataset. We compare our methods with the following Laplace-based algorithms (details in Appendix B): *NeuralLinear* [Snoek et al., 2015, Riquelme et al., 2018], *PrecisionDiag* [Riquelme et al., 2018, Zhang et al., 2020], *Subnet diagonal* [Daxberger et al., 2021b]. For thoroughness, we include in our comparison, two standard (non-Laplace) Bayesian methods, the *Split* method [Kuchibhotla et al., 2023] and the *ensemble* method [Lakshminarayanan et al., 2017, Osband et al., 2016].

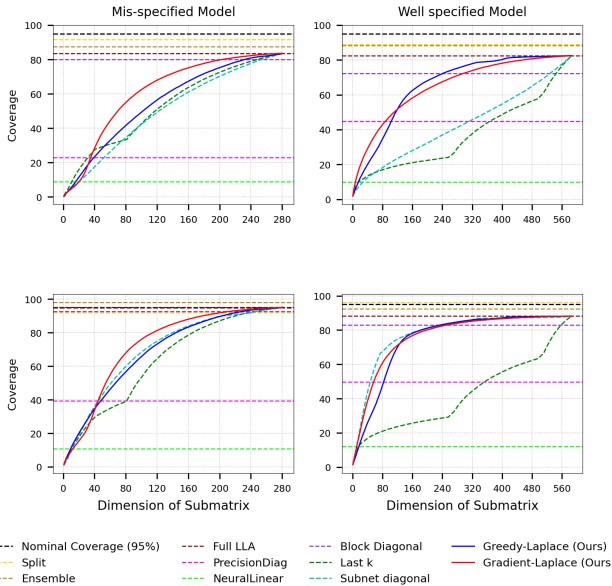

Figure 2: For various methods, coverages of the credible intervals using the full Laplace posterior, the relevant surrogate Laplace posterior, the split and the ensemble methods averaged over the entire test data set, and then over 200 replications are depicted. The x-axis corresponds to the subset size $k$ (for relevant subnetwork LA methods), and the y-axis corresponds to the empirical coverage described above. For both regression and classification, two working model settings are considered. The top row corresponds to the **regression** setup, and the bottom row corresponds to the **classification** setup. The plots in left column are corresponding to the mis-specified working model setting, and those in the right column are corresponding to the well-specified working model setting.

The *linearized Laplace approximation (LLA)* provides overall competitive coverage to the two benchmark methods, *Split* and *Ensemble*. Among the methods that use a low-dimensional surrogate of the Hessian to further approximate LLA, *NeuralLinear* clearly provides poor coverage. Its immediate extension, *Last k*, also provides much lower coverage compared *Subnet diagonal*, and both the proposed methods, *Greedy-Laplace* and *Gradient-Laplace*, especially in the well-specified working model settings. The proposed *Gradient-Laplace* approach clearly outperforms the *Subnet diagonal* approach in all but the well-specified working model setting under classification, where the *Subnet diagonal* provides marginally better coverage. The proposed *Greedy-Laplace* approach provides similar coverage to the *Subnet diagonal* approach in mis-specified working model settings, and significantly outperforms it in the well-specified working model setting for the regression setup, while the *Subnet diagonal* approach provides marginally better coverage in the well-specified setting for the classification setup. Although *PrecisionDiag* provides better coverage than *NeuralLinear*, the coverage is still quite low, and is attained by the proposed methods for very small values of $k$. To summarize, the proposed methods for subnetwork LA overall provide significantly better coverage than existing methods for constructing surrogates for the Laplace precision matrix.

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

## A   Additional Experimental Details for Section 4.1

We generate synthetic regression data, with 2000 samples, using 5 input variables and a single output variable. The input variable $\mathbf{x}$ is generated from $N_5(\mathbf{0}, \Sigma)$, where $\Sigma_{i,j} = 0.6^{|i-j|}$, $i, j = 1, 2, \ldots, 5$, and a 3-hidden layer fully connected feed-forward Neural Network with random weights is used to generate the output. The error variance $\sigma_0^2$ is fixed at 1. In the **mis-specified setting**, the data generating model is a 3-hidden layer Neural network with 30 nodes in each layer, and $tanh$, $sigmoid$ and $tanh$ activation functions respectively, and the working/fitted model is again 3-hidden layer Neural network with 10 nodes in each layer, and $tanh$, $sigmoid$ and $tanh$ activations respectively. We use a 3-hidden layer Neural network with 15 nodes in each layer, and $tanh$, $sigmoid$ and $tanh$ activation functions respectively, as both the true model and the working model in the **well-specified setting**. We generate 200 training data for the study. 10% of the training data has been used as validation data. The in-distribution, OOD and mixed test data are all independently generated from $N(\mathbf{1}_5, \mathbf{I}_5)$, $N(6\mathbf{1}_5, \mathbf{I}_5)$ and $N(\mathbf{0}, 25\mathbf{I}_5)$ respectively. We have 500 test data points in each category, and the Wasserstein based metric for each algorithm is computed by averaging the relevant Wasserstein distance over the 500 data points.

We use AdamW for optimization. To train the models, we consider IID $N(0, 1)$ priors on the weights and biases, and the noise variance $\sigma_0^2$ to be 1 (the true value). We use a decaying learning rate starting at 0.01, and after every 500 epochs, it is multiplied by 0.1 and trained the model over 1500 epochs.

## B   Benchmark Algorithms

We compare the proposed methods with the following benchmark algorithms.

- **NeuralLinear** [Snoek et al., 2015, Riquelme et al., 2018] is a subnetwork LA based approach where $S$ is chosen to be the subset of parameters associated with the last layer of the neural network. Its natural extension, the **Last k** approach, arranges the parameters in order starting from the first layer of the network to the last layer of the network, and for a given subset size $k$, chooses the last $k$ parameters in this order as the elements of $S$.

- **PresicionDiag** [Riquelme et al., 2018, Zhang et al., 2020] approximates the Laplace precision matrix with its diagonal matrix, setting all the off-diagonals to zero.

- **Subnet diagonal** [Daxberger et al., 2021b] is another sub-network LA approach, which for a given subset size $k$, selects the parameters associated to the $k$ smallest diagonal entries of $\Omega$ as the elements of $S$.

- **Block Laplace** uses a block diagonal surrogate for $\Omega$ as an alternative to the PrecisionDiag approach in [Riquelme et al., 2018]. In particular, parameters are partitioned into blocks based on which layer of the neural network they belong to, and the corresponding block diagonal version of $\Omega$ is used as a replacement for $\Omega$ in (1).

- **Split** method [Kuchibhotla et al., 2023] splits the training data into $d$ parts, uses each of them to fit the model separately and then proposes the highest and the lowest estimated value of $f_{\boldsymbol{\theta}}(\mathbf{x})$ among the $d$ fitted models as the upper and lower limits of the confidence interval for $f_{\boldsymbol{\theta}}(\mathbf{x})$ respectively.

- **ensemble** method [Lakshminarayanan et al., 2017, Osband et al., 2016] generates $B$ bootstrap samples using the training data, uses them to train the network separately, and then proposes the 2.5% and 97.5% quantiles to be the lower and upper limit respectively of a 95% confidence interval for $f_{\boldsymbol{\theta}}(\mathbf{x})$.

## C   Linearized Laplace for Classification

The posterior predictive distribution for a new observation $(\mathbf{x}^*, y^*)$ in the binary classification setup will be $P(y^* = 1 | \mathbf{x}^*, \mathcal{D}_{train}) = \sigma(\mu(\mathbf{x}^*))$, where $\mu(\mathbf{x}^*) \sim N(f_{\hat{\boldsymbol{\theta}}_{MAP}}(\mathbf{x}^*), \sigma_0^2 + g(\mathbf{x}^*)^T \Omega^{-1} g(\mathbf{x}^*))$, $\sigma(\cdot)$ is the sigmoid function, and

$$\Omega = \sum_{n=1}^{N} g(\mathbf{x}_n) \left( \nabla_f^2 \log p(\mathbf{y}_n | f)|_{f = f_{\hat{\boldsymbol{\theta}}_{MAP}}(\mathbf{x}_n)} \right) g(\mathbf{x}_n)^T + (\text{diag}(\mathbf{v}))^{-1} \tag{C.1}$$

with $\nabla_f^2 \log p(\mathbf{y}_n | f) = e^f / (1 + e^f)^2$.

## D    Additional Experimental Details for Section 4.2

**Data Structure and Data Generation.** Under each setting, we generate the synthetic data with 5000 sample points, each with 5 input variables and a single output variable for our experiment. The input variable $\mathbf{x}$ is generated similarly as in Section 4. A fully connected feed-forward Neural Network is used to compute the outcome variable. For both classification and regression experiments, the true model used to generate the outcome corresponding to the mis-specified setting is a neural network with 3 hidden layers, 25 hidden nodes in each layer and $tanh$, $sigmoid$ and $tanh$ activation functions respectively, whereas the true model (and the working model) in the well-specified setting is a 3-hidden layer Neural network with $tanh$, $sigmoid$ and $tanh$ activations again, but with 15 hidden nodes in each layer. The weights and biases of the data generating neural network are generated independently from the standard normal distribution. In the regression setup, the response is Gaussian with variance 1, whereas in classification, the binary response is generated from a Bernoulli distribution. The test data is generated independently from the $N(0, 25)$ distribution.

We use AdamW for optimization. $10\%$ of the training data is used as validation data, and the learning rate and number of iterations is chosen by observing the behaviour of training and validation loss of 5 randomly selected datasets out of 200. For the split method, we split the test data into $d = 8$ parts, and adjust the learning rate accordingly (since, the size of the training data for each training decreases), and for the ensemble method, we generate $B = 20$ bootstrap samples, each of size 5000, and fit 20 models using those samples.

As mentioned earlier, for both the regression and classification experiments, we have two working model settings. For both these settings, we consider IID $N(0, 1)$ priors on the weights and biases. For the **mis-specified setting**, the working model is a Neural network with a single hidden layer with 40 hidden nodes and $tanh$ activation function. For the **well-specified setting**, as previously mentioned, the working model has 15 hidden nodes in all 3 hidden layers and $tanh$, $sigmoid$ and $tanh$ activations respectively. To train the models, we use a decaying learning rate to train the models. For regression experiments, the initial learning rate is 0.01 for both the working model settings, whereas for classification experiments, the learning rate is 0.01 for the mis-specified setting, and 0.001 for the well-specified setting. In all 4 cases, the learning rate is multiplied by 0.1 after every 1000 epochs. We use 2000 epochs to train each of the models.

Once the network is trained, we compute the empirical coverage of the credible interval of $f_{\boldsymbol{\theta}}(\mathbf{x})$ for all the benchmark approximation methods, as well as the proposed methods. The coverages are further averaged over 200 replications. The results are depicted in Figure 2.

