# OpenReview forum: "Computationally Efficient Laplace Approximations for Neural Networks"
_NeurIPS.cc/2024/Workshop/BDU — NeurIPS BDU Workshop 2024 Poster_

### Official Review · Reviewer_gvpQ · 2024-09-26

**Rating:** 6
**Confidence:** 3

**Review:**

This paper proposes two novel methods for approximating the Hessian matrix in Laplace approximations for neural networks: Greedy-Laplace and Gradient-Laplace. These methods aim to address the computational bottlenecks associated with the inversion of large Hessian matrices by selecting subsets of parameters that contribute the most to the posterior variance. The paper presents a clear motivation for sparse Hessian approximations and provides empirical evidence on synthetic datasets showing that the proposed methods outperform existing subnetwork-based Laplace approximations. The topic is timely and relevant to the workshop, the use of Laplace approximations is well-grounded in Bayesian statistics, and efforts to make these methods computationally tractable are both timely and valuable.
My comments are as follows:
1.	How to consider the trade-offs between computational complexity and the quality of the approximation (in terms of posterior variance)?
2.	Are there formal error bounds for the approximation using the Moore-Penrose inverse versus the true inverse of the precision matrix?
3.	Additionally, while the Greedy-Laplace algorithm intuitively selects parameters based on the smallest diagonal entries, it would be valuable to discuss the theoretical properties of this subset selection approach in more detail, such as whether this strategy can lead to suboptimal selections in some cases.
4.	The paper demonstrates improvements in computational efficiency, however, the current experiments are primarily limited to synthetic data with a relatively small number of parameters (5 inputs and 3 hidden layers). To claim practical relevance, it would be important to test the methods on real-world datasets and/or larger neural network architectures (e.g., convolutional neural networks or transformers).
5.	I’m wondering: Can the proposed methods scale to models with millions of parameters?
6.	How does the runtime compare to other efficient uncertainty quantification methods? Please discuss.
7.	How do Greedy-Laplace and Gradient-Laplace compare to popular Bayesian neural network approaches (e.g., variational inference or MCMC-based methods)?
8.	When Gradient-Laplace is computationally preferable (versus the Greedy-Laplace)? I think adding more details on this point would help practitioners understand when to use each method.\

---

### Official Review · Reviewer_v38U · 2024-10-07
**This paper introduces two innovative methods for improving computational efficiency in uncertainty quantification for neural networks, with strong performance and relevance in various applications, but it could benefit from clearer explanations and more real-world validation**

**Rating:** 6
**Confidence:** 3

**Review:**

Pros:
This paper Introduces Greedy-Laplace and Gradient-Laplace, improving computational efficiency for uncertainty quantification.
It reduces computational complexity, making it feasible for large neural networks.
It has Rigorous benchmarking against state-of-the-art methods using multiple datasets.
It addresses important applications like healthcare and autonomous systems.
The outperforms existing methods in terms of accuracy and credible interval coverage.

Cons:
Its complex mathematical details may be difficult for non-experts.
It lacks extensive real-world application demonstrations.
The minimal exploration of where the methods might fail.
And the gradient-based method may introduce computational overhead.

In summary, the paper presents an important contribution to computationally efficient uncertainty quantification in neural networks. Its originality, practical relevance, and rigorous evaluation make it a valuable piece of research in the field of Bayesian neural networks and Laplace approximations. However, further clarity in presentation and real-world validation would enhance its impact.

---

### Decision · Program_Chairs · 2024-10-09

Accept (Poster)